# A Novel Dual-Polarized Magnetoelectric Dipole Antenna and Its Array for LTE and 5G Sub-6 GHz Base Station Applications

**DOI:** 10.3390/e25020274

**Published:** 2023-02-02

**Authors:** Zhiwei Song, Junwu Qi

**Affiliations:** 1School of Electrical Engineering, Xinjiang University, Urumqi 830046, China; 2The Unisoc (Shanghai) Technologies Co., Ltd., Shanghai 201203, China

**Keywords:** dual-polarized, magnetoelectric dipole, base station antenna, antenna array

## Abstract

This study presents a novel dual-polarized magnetoelectric dipole antenna and its array with director and rectangular parasitic metal patches for LTE and 5G sub-6 GHz base station applications. This antenna is composed of L-shaped magnetic dipoles, planar electric dipoles, rectangular director, rectangular parasitic metal patches, and η-shaped feed probes. The gain and bandwidth were enhanced by using the director and parasitic metal patches. The measured impedance bandwidth of the antenna was 82.8% (1.62–3.91 GHz, VSWR < 1.5), and its gain was 10 ± 0.5 dBi. The profile of the antenna unit, operated at 1.7 GHz, was only 42 mm (0.227*λ*_0_, where *λ*_0_ represents the free space wavelength corresponding to the lowest resonance frequency point). Subsequently, four antenna units were arranged in a line array with 0.6*λ*_0_ spacing. Both the antenna and its array were fabricated and measured. The measurement results show that the array has good radiation characteristics, such as broad bandwidth covering 1.65–3.97 GHz (VSWR < 1.5), high gain (its gain was great than 15.2 dBi), and high radiation efficiency (>90%). Its HPBWs were 63° ± 4° and 15° ± 2° for H- and E-planes, respectively. The design can cover TD-LTE and 5G sub-6 GHz NR n78 frequency bands very well, meaning that this is a good candidate antenna for base station applications.

## 1. Introduction

Broadband dual-polarized antennas have been widely used in base stations due to their merits, such as high channel capacity, superior resolution, and precise ranging [1]. With the rapid development of mobile communication technology, wideband antennas are required to cover multiple communication bands such as GSM1800, CDMA1900, TD-SCDMA, and LTE systems (1.71–2.69 GHz, 44.5%) [1,2,3,4,5,6,7,8]. In [1], the bandwidth, gain, and dimensions of a magnetoelectric (ME) dipole antenna, a dipole antenna, and a microstrip antenna are compared, and it is clear that the first kind of antenna has better characteristics. In [3], bandwidth of the antenna is 1.32–2.74 GHz, covering 2G/3G/4G bands. In [4], a low-profile dual-polarized omnidirectional antenna is designed, and the isolation of the antenna is around 35 dB. In [5], the impedance bandwidth of the antenna covers 1.39–2.76 GHz, and the side-lobe level is greater than 16 dB. In [8], the design is two bands working cover 0.69–0.96 GHz and 1.69–2.69 GHz, respectively.

Recently, the sub-6 GHz band (3.30–3.60 GHz) has received significant attention due to its value to 5G services, which can support a faster communication rate. To support all the systems mentioned above, its bandwidth must be more than 71.2% (i.e., 1.71–3.60 GHz). Consequently, there is a need for dual-polarization base station antennas with good performance in broadband [9,10,11,12,13,14,15,16].

In this paper we propose a dual-polarized magnetoelectric dipole antenna array covering GSM1800, CDMA1900, TD-SCDMA, WIFI, WiMax, TD-LTE (2.3–2.7 GHz), and 5G sub-6 GHz NR n78 (3.3–3.8 GHz) frequency bands, realizing the more stable gain in a wideband. Four antenna units are linearly arranged with a space of about 0.6*λ*_0_ between adjacent units. Four horizontal electric dipoles and four vertical magnetic dipoles are arranged symmetrically around the *z*-axis and excited simultaneously by two η-shape feeders to gain dual-polarized radiation for each unit. To stabilize the radiation pattern and enhance its gain with a half-power beam width (HPBW) of 62° ± 4°, a director and four parasitic metal patches are introduced into the ME dipole antenna unit. The dimensions and spacing of the director and parasitic metal patches provide more degrees of freedom for gain adjustments. The gain is enhanced by using the director and parasitic patches.

## 2. Topology and Design Strategy

### 2.1. Geometry of the Antenna and ME Dipoles

The 3D view, side view, and top view of the proposed ME dipole antenna unit are shown in Figure 1 and the specific parameters are listed in Table 1. The designed unit is symmetrically arranged around the *z*-axis and primarily consists of seven parts: four horizontal electric dipoles, four vertical magnetic dipoles, a director, four square metal parasitic patches, two η-shape feed probes, an open square box reflector, and plastic fasteners. The exploded view of the proposed ME dipole antenna element and corresponding parameters are shown in Figure 2. The ME dipole antenna elements, reflector, parasitic patches, director, and η-shape feeders are made of 0.5 mm-thickness copper; therefore, the antenna has a strong mechanical structure and is lightweight. To reduce assembly error and reinforced antenna structure as much as possible, 3D printed plastic fasteners are used. Their influence on the radiation characteristics of the antenna is negligible, therefore no detailed parameter description is given for the sake of brevity.

In the simulations, plastic fasteners are modeled as white ABS-M30 cylinders with different diameter settings (mounting parts are 1.0 mm, and conversion connection parts are 3.0 mm). The positions of the mounting holes are shown in Figure 2. The total profile of the unit is 0.227*λ*_0_ (42.0 mm), corresponding to the free space wavelength at 1.7 GHz. The feeders are composed of one horizontal plane and two vertical planes, as shown in Figure 2d,e. The longer part of Feeder 2 is shorter than the longer part of Feeder 1 by about 3.5 mm. The Feeders are perpendicular to the horizontal reflector and with a gap of 1.7 mm, where the feed points are connected to the SMA connector.

The simulated current distributions on the radiation copper patch surfaces at 1.7, 2.4, 3.1, and 3.7 GHz are shown in Figure 3. The four patches on the top are working in the same manner as electric dipoles. The L-shape patch and the top patch construct a special structure and work similarly to an open loop; by doing so they form the magnetic dipoles together. The combination of symmetrical magnetoelectric dipoles makes the element obtain dual polarization characteristics. From Figure 3a, the length of the effective current path should be *e_y_* + *m_z_* + *m_y_* + *p* = 85 mm at 1.7 GHz, which is about half of the free space wavelength (0.48*λ*_0_) corresponding to this frequency point. In other words, the actual radiation unit is composed of a horizontal radiation patch, an L-shaped vertical radiation patch, and a parasitic patch. From Figure 3b, the length of the effective current path should be *e_y_* = 30 mm at 2.4 GHz, which is about 1/4 (0.24*λ*_0_) of the free space wavelength corresponding to this frequency point. According to the above analysis and the simulation results in Figure 3, the maximum current distribution at 1.7 and 2.4 GHz appear on the vertical and horizontal plane, respectively. Therefore, the antenna element at the low-frequency points is similar to the traditional magnetoelectric dipole. At 3.1 and 3.7 GHz, the nulls appear near the end of the horizontal patch and the current distributions on the vertical plates are irregular. It can be seen that the magnetoelectric dipole antenna unit at the high-frequency points works in the high-order mode, and the calculation results are close to the simulation results, as shown in Figure 3.

### 2.2. Structure Evolution Process of the ME Dipole Antenna

The evolutionary process of the antenna element geometry is shown in Figure 4. The simulated VSWRs and gains using HFSS 21.0 of different antennas are shown in Figure 5. The maximum surface current distributions of different antennas at 1.7 GHz are shown in Figure 6.

**Case 1:** The VSWR is almost greater than 2 in the 1.69–3.81 GHz band, and the gain fluctuates greatly at the same time.

**Case 2:** The impedance matching is better when the director is used, and the VSWR is less than 1.5 from 2.4 to 3.9 GHz. However, the gain is smaller and fluctuates greatly.

**Case 3:** The VSWR is less than 1.5 from 1.6 to 3.9 GHz, its impedance bandwidth covers the design goals very well, and its gain is more stable. The variation range of gain is 0.7 dB in the 1.71–2.69 GHz frequency band, and 1.0 dB in the 3.3–3.6 GHz frequency band, respectively.

From Figure 6, we can see that the maximum surface current of Case 2 is greater than Case 1 when the director is used in the antenna unit. Additionally, the maximum surface current of Case 3 is the greatest of the aforementioned cases, due to both the director and parasitic metal patches being used. This phenomenon shows that the proposed antenna structure can enhance the gain of the antenna element and improve the radiation efficiency of the antenna by concentrating more energy in the ME dipoles.

The simulated input impedance and S_21_ of the three cases are shown in Figure 7. Both the real and imaginary parts of Case 3 are more stable than the other two cases by using the proposed geometry. The three lines without symbols at the bottom of Figure 7 are simulated S_21_. It is clear that the proposed antenna with good isolation is between ports 1 and 2, and there is less than −27 dB in the whole operating frequency band.

### 2.3. Working Principle of the Magnetoelectric Dipole Antenna Unit

It can be seen from Figure 1 that the magnetoelectric dipole antenna unit consists of a reflector, excitation source, radiation patches, parasitic patches, and a director from bottom to top. Therefore, its structure is similar to the Yagi antenna. A director (square copper patch) is installed above the horizontal copper patches (radiation patches and parasitic patches). The horizontal copper patches work as a driven system and will excite the director within the LTE and 5G Sub-6 GHz frequency bands. Since the working mode of the magnetoelectric dipole antenna unit is similar to a quasi-Yagi antenna, its analysis method can refer to Reference [1]. Next, we will focus on the influence of parasitic patches on the radiation performance of the antenna unit. The current distribution of radiation and parasitic patches simulations at 1.7 GHz is shown in Figure 8. The components of the current vectors on the *x* and *y* axes of the parasitic patches are consistent with the components of the current vectors on the *x* and *y* axes of the radiation patches, respectively. Therefore, the electric field strength on the radiation patches is enhanced after adding the parasitic patches. There are similar trends at other frequency points. For further explanation, the variation trends of electric field components on the *x*-axis and *y*-axis under different conditions are shown in Figure 9, wherein solid lines correspond to Case 2, and signed lines correspond to Case 3. The electric field strength of the radiation patches has been enhanced, which means that the antenna gain and VSWR are improved, as shown in Figure 5.

### 2.4. Simulation Analysis of Key Parameters of the ME Dipole Antenna

Many key parameters influence the proposed ME antenna performance greatly. We give some simulated results of reflection coefficients in the following section for further explanation. The simulated reflect coefficients are shown in Figure 10, when ME dipole width (*e_x_*, lines with hollow symbols), magnetic dipole height (*m_z_*, lines with solid symbols), or electric dipole length (*e_y_*, lines without symbols) are with different sizes. The simulated S11 of the proposed ME dipole antenna has 3 operating frequency points as shown in Figure 10, and they are 1.7, 2.4, and 3.7 GHz; their corresponding *λ*_0_/4 wavelengths are 44.1, 31.2, and 20.1 mm, respectively. The sum of magnetic or electric dipole width and length is 44.0 mm, the lengths of them are 30.0 mm, and the widths of them are 14.0 mm, corresponding to 3 operating frequency points. Therefore, all the changes in the above-mentioned parameters have a great influence on the S_11_ in the whole operating frequency band. We obtain the desired operating frequency by adjusting the size of these parameters in the design stage.

The simulated S_11_ when the size of feeder 1’s width (*f*_1*x*1_, *f*_1*x*2_), or spacing between director and electric dipoles (*H*) is changed, is shown in Figure 11. It is clear that the size change of these parameters has little influence on the bandwidth of the ME antenna; however, this can affect the input impedance greatly. Thus, we can gain good input impedance by changing these sizes.

## 3. Measurement Results of the Antenna Unit

We use an Agilent vector network analyzer (PNA-X) and a microwave anechoic chamber (OBT-6) to measure the proposed ME antenna unit. The measured and simulated reflect coefficients are shown in Figure 12. When the S_11_ < −15 dB, the simulated S_11_ covers 1.57–3.94 GHz frequency bands and the fractional bandwidth is 86.0%; when the measured S_11_ covers 1.62–3.91 GHz frequency bands, the fractional bandwidth is 82.8%. The results show that the proposed antenna has wide bandwidth and covers the design goals very well. The simulated and measured port-to-port isolations are greater than 27.2 dB, and the experimental results are consistent with the simulation results. The simulated and measured radiation patterns at 1.7, 2.4, 3.2, and 3.7 GHz are shown in Figure 13. The radiation patterns have good symmetry in the whole operating frequency band due to the antenna unit being symmetrical about the *z*-axis. The HPBW is from −35° to 30° for port 1, and from −30° to 34° for port 2, at 1.7, 2.4, and 3.1 GHz. The HPBW is from −29° to 33° for port 1, and from −33° to 31° for port 2, at 3.7 GHz. In addition, the front-to-back ratio (FBR) is greater than 20 dB in the entire operating frequency band. The simulated and measured gain is shown in Figure 14. The antenna unit has stable gain in the whole frequency band, and the gain is great than 9.5 dBi. The measured radiation efficiency is also shown in Figure 14, and it is greater than 91% in the operating frequency band.

The changes in other parameters have little effect on the antenna reflection coefficient, therefore they are not listed for the sake of brevity.

## 4. Antenna Array

The photo of the fabricated prototype of the proposed antenna array is shown in Figure 15. The antenna elements in the presented antenna array seem to be twisted slightly for a few degrees, due to the parasitic copper sheets being clamped between the upper and lower plastic fasteners and installed on the antenna unit. The plastic fasteners made by 3D printing technology are only 0.5 mm thick, therefore, the rigidity of the plastic support is limited, making its surface not flat, and causing some distortion of the antenna unit. The comparison between the measured data and the simulation data in the following section shows that the radiation characteristics of the antenna array will not change significantly due to this deformation. From another point of view, this antenna array structure has high robustness. The antenna array is composed of four ME antenna units, and the spacing between two adjacent units is 0.594*λ*_0_ (110 mm). The measured reflect coefficients and gain are shown in Figure 16. Both the S_11_ and S_22_ cover the 1.65–3.97 GHz, fractional bandwidth is 82.7%, and this matches our design goals very well, as mentioned above. The measured S_21_ is less than −30 dB in the entire operating frequency band, due to the optimized spacing between adjacent units. The measured and simulated normalized radiation patterns of the antenna array at 1.7, 2.4, 3.2, and 3.7 GHz are shown in Figure 17. The FBR is greater than 20 dB at 1.7 GHz, and is greater than 25 dB at 2.4, 3.2, and 3.7 GHz. In addition, the radiation patterns of the proposed antenna array have good symmetry in the whole operating frequency band. For phi = 0°, the HPBWs are 66.6°, 65.6°, 59.8°, and 64.6° at 1.7, 2.4, 3.2, and 3.7 GHz, respectively. For phi = 90°, the HPBWs are 19.4°, 14.2°, 11.1°, and 9.4° at 1.7, 2.4, 3.2, and 3.7 GHz, respectively. The co-polarization is 25 dB larger than the cross-polarization in the entire operating frequency band.

To better address the advantages of the proposed ME antenna unit, comparisons with references are listed in Table 2, including dimensions, bandwidth, radiation efficiency, and gain. The gain of our design is the best one. We use the higher standard of *−*15 dB, and the bandwidth reaches 82.8%. Although the size of the antenna we designed is not the smallest, there is little difference compared with the design in the references. The proposed antenna is a competitive candidate for base-station applications due to its merits, such as stable radiation patterns, flat gain, and high FBR.

## 5. Conclusions

In this paper, a broadband dual-polarized ME dipole antenna and a 4-units array have been proposed and fabricated for LTE and 5G sub-6 GHz applications. A broadband ME dipole antenna is obtained by using director and parasitic metal patches. Moreover, the magnetoelectric dipole elements are placed symmetrically about the *z*-axis, and a symmetrical radiation pattern is obtained. As the measured results indicate, the antenna unit has an operating bandwidth from 1.62 to 3.91 GHz (82.8%) with VSWR < 1.5 and isolation > 27 dB. The antenna has an HPBW within 63° ± 2° and a gain within 10.0 ± 0.5 dBi in the entire operating frequency band. As the measured results indicate, the antenna array has an operating bandwidth from 1.65 to 3.97 GHz (82.7%) with VSWR < 1.5 and isolation > 32 dB. The array has an HPBW within 63° ± 4° and a gain greater than 15.2 dBi in the entire operating frequency band. They are good candidates for base station use.

## Figures and Tables

**Figure 1 entropy-25-00274-f001:**
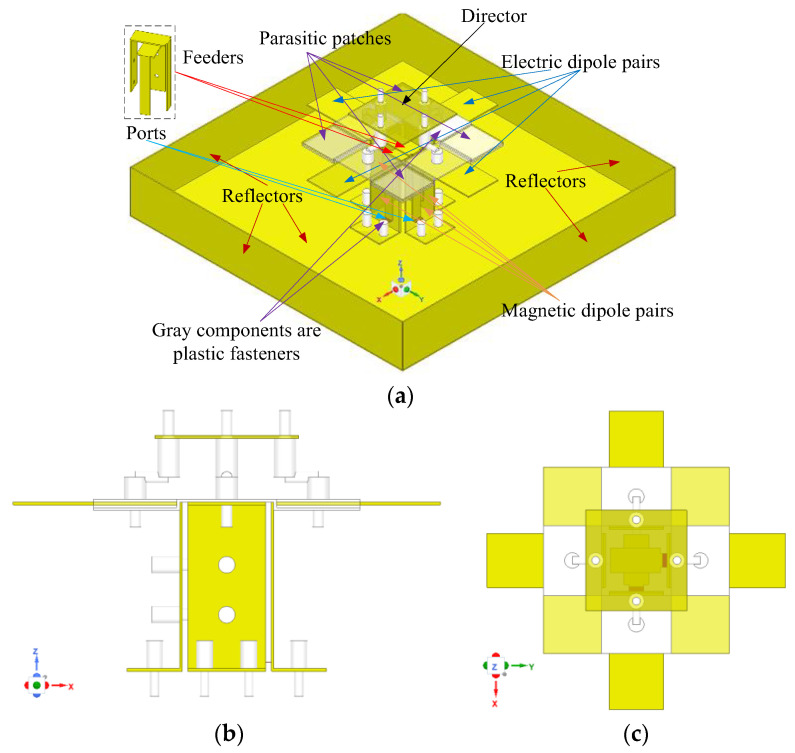
3D configuration of proposed ME dipole antenna unit. (**a**) is a 3D view of the antenna unit, and (**b**,**c**) are the side view and top view of the antenna unit without the reflector, respectively.

**Figure 2 entropy-25-00274-f002:**
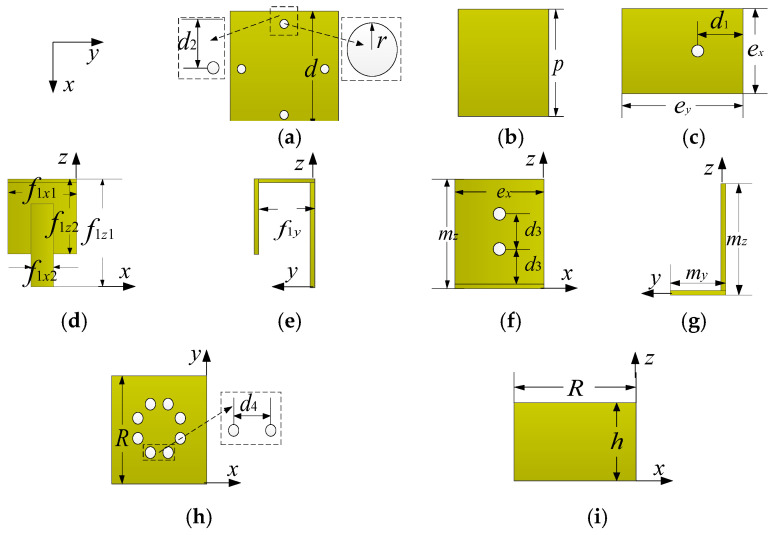
Exploded view of proposed ME dipole antenna element and corresponding parameters. (**a**) is the top view of the director; (**b**) is the top view of the parasitic patch; (**c**) is the top view of the electric dipole unit, which are parallel to the *x*-*z* plane; (**d**,**e**) is the front and side view of feeder 1; (**f**,**g**) are the front view and side view of the magnetic dipole unit; and (**h**,**i**) are the top and side view of the reflectors. The structures of feeder 1 and feeder 2 are similar, so only feeder 1 is shown. The thickness of all copper plates is 0.5 mm, so they are not marked one by one in the figure. The coordinate system of each subgraph in Figure 2 is consistent with that in Figure 1.

**Figure 3 entropy-25-00274-f003:**
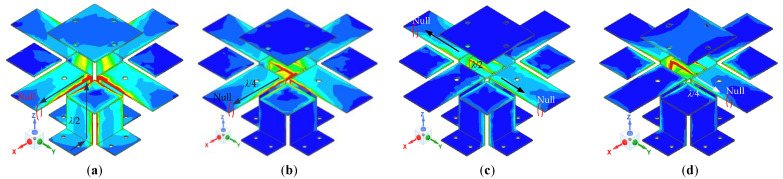
Current distributions on the radiation patches at (**a**) 1.7 GHz, (**b**) 2.4 GHz, (**c**) 3.1 GHz, and (**d**) 3.7 GHz. The unit of surface current is A/m.

**Figure 4 entropy-25-00274-f004:**
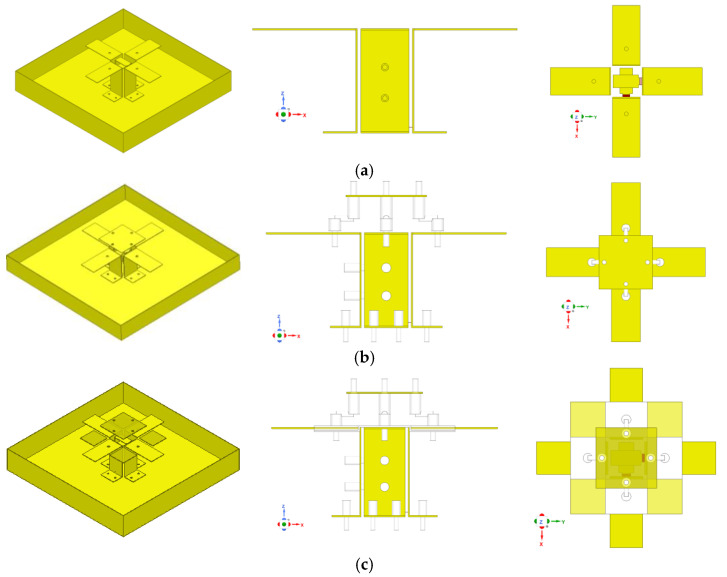
The configuration evolution of the designed ME dipole antenna. (**a**) ME dipoles only (case 1). (**b**) ME dipoles with the director (case 2). (**c**) Proposed antenna (case 3).

**Figure 5 entropy-25-00274-f005:**
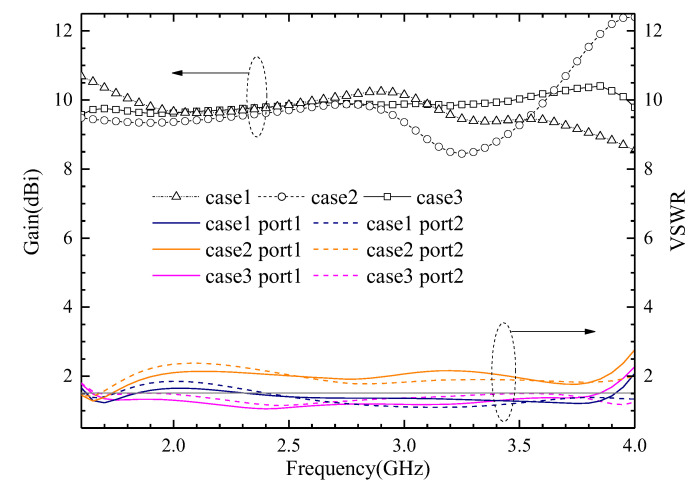
The simulated VSWRs and gains of the three cases.

**Figure 6 entropy-25-00274-f006:**
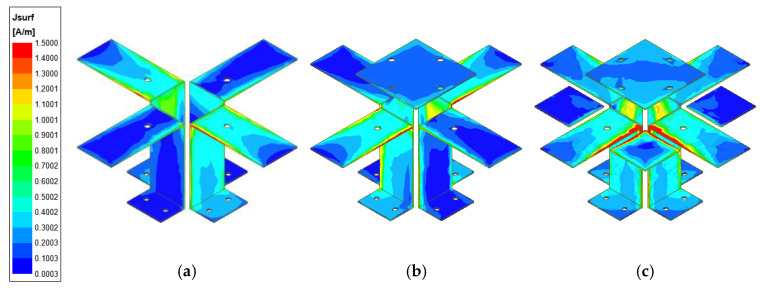
The simulated maximum surface current distributions of three different antennas at 1.7 GHz. (**a**) Case 1. (**b**) Case 2. (**c**) Case 3. The unit of surface current is A/m.

**Figure 7 entropy-25-00274-f007:**
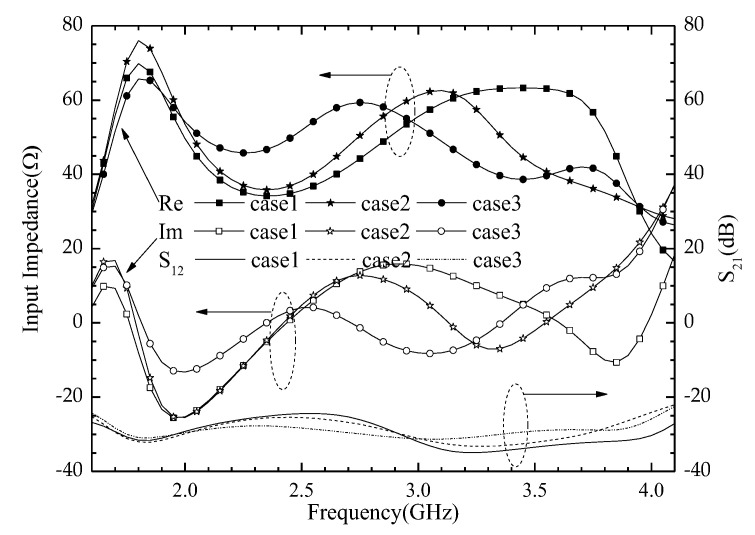
The simulated input impedance and S_21_ of three cases.

**Figure 8 entropy-25-00274-f008:**
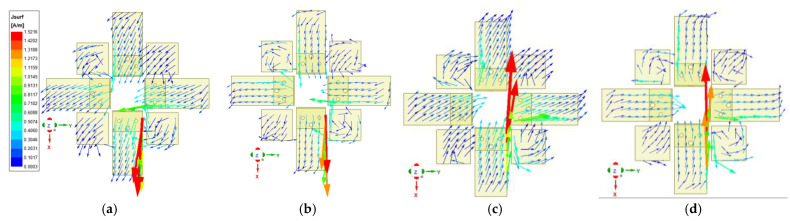
The simulated current distribution of the radiation and parasitic patches at 1.7 GHz. (**a**) t = 0, (**b**) t = T/4, (**c**) t = T/2, (**d**) t = 3T/4.

**Figure 9 entropy-25-00274-f009:**
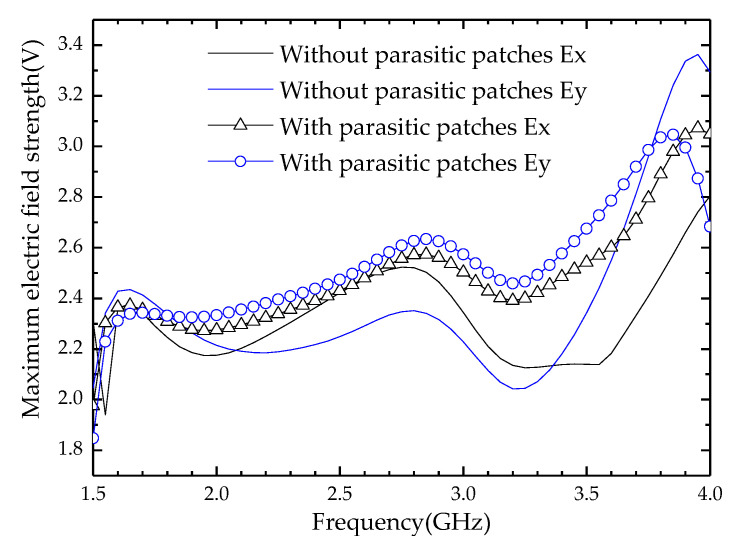
The simulated maximum electric field strength at the whole working frequency band.

**Figure 10 entropy-25-00274-f010:**
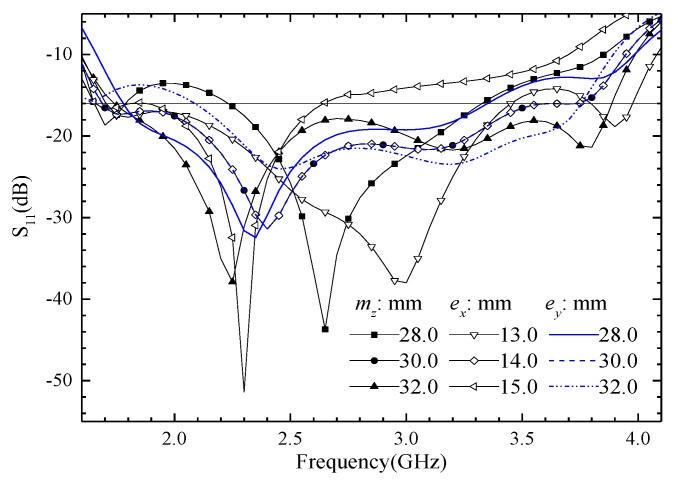
The simulated S_11_ is when the size of ME dipole width (*e_x_*), magnetic dipole height (*m_z_*), or electric dipole length (*e_y_*) is changed.

**Figure 11 entropy-25-00274-f011:**
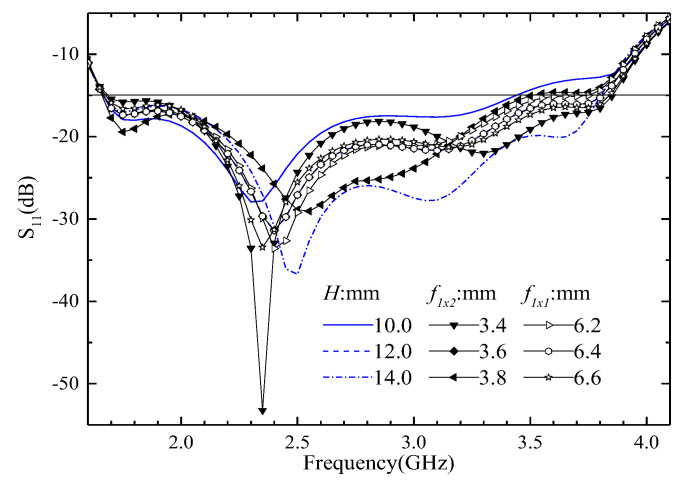
The simulated S_11_ when the size of feeder 1 width (*f*_1*x*1_, *f*_1*x*2_), or distance between director and electric dipoles (*H*) is changed.

**Figure 12 entropy-25-00274-f012:**
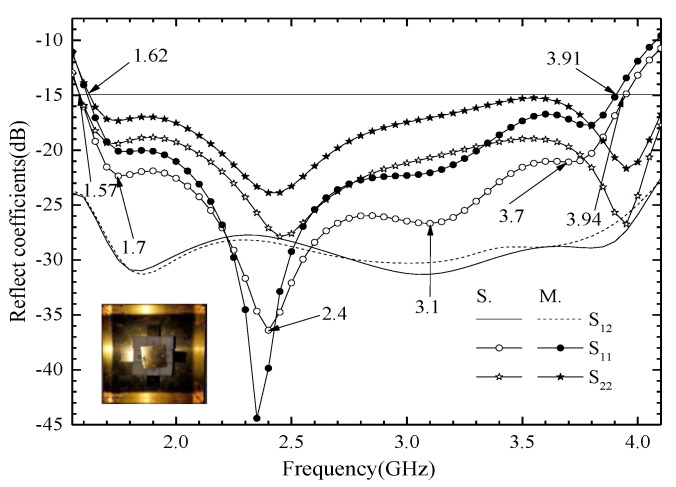
Simulated and measured reflection coefficients of the proposed ME antenna unit.

**Figure 13 entropy-25-00274-f013:**
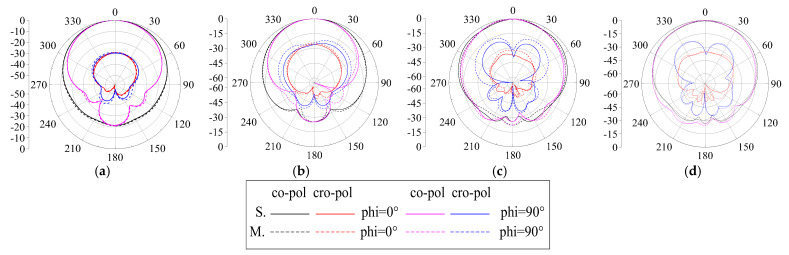
Simulated and measured radiation patterns of the proposed unit. (**a**) 1.7 GHz. (**b**) 2.4 GHz. (**c**) 3.2 GHz. (**d**) 3.7 GHz.

**Figure 14 entropy-25-00274-f014:**
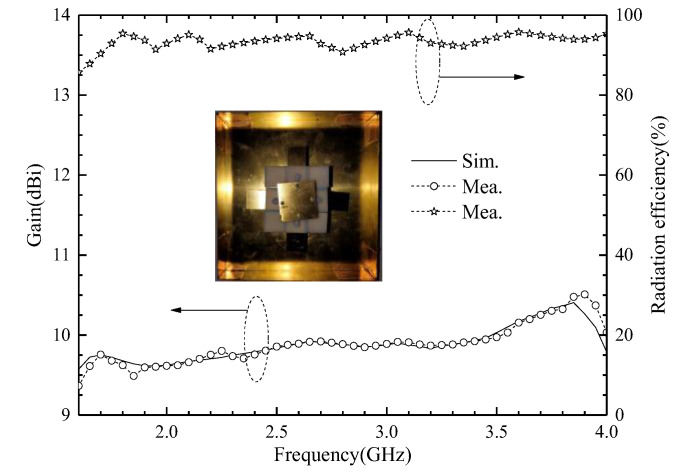
Simulated and measured gains of the proposed ME antenna unit, and its measured radiation efficiency.

**Figure 15 entropy-25-00274-f015:**
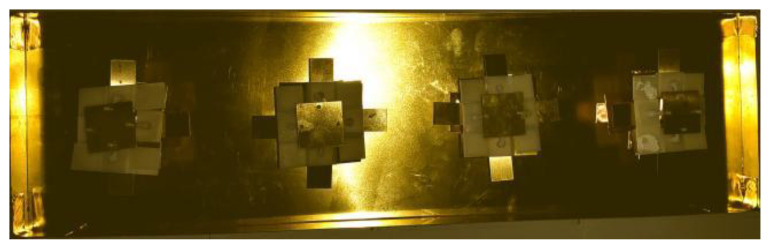
Fabricated prototype of the proposed ME antenna array.

**Figure 16 entropy-25-00274-f016:**
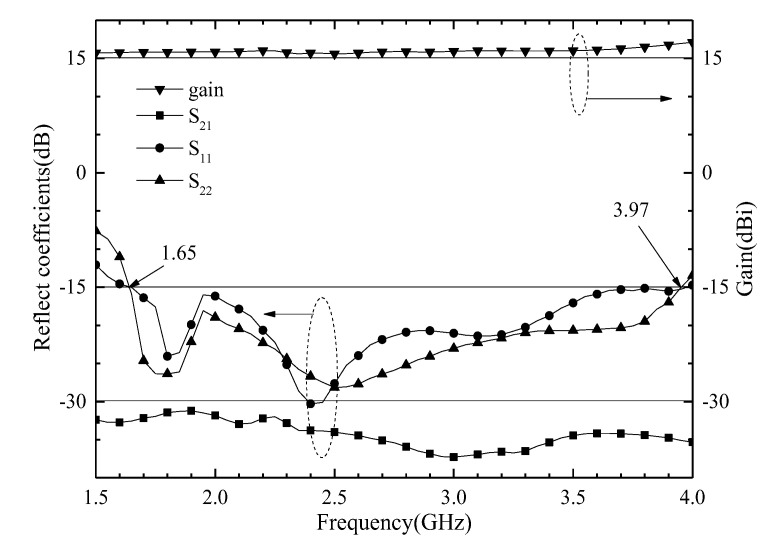
Measured reflect coefficients and gain of the proposed ME antenna array.

**Figure 17 entropy-25-00274-f017:**
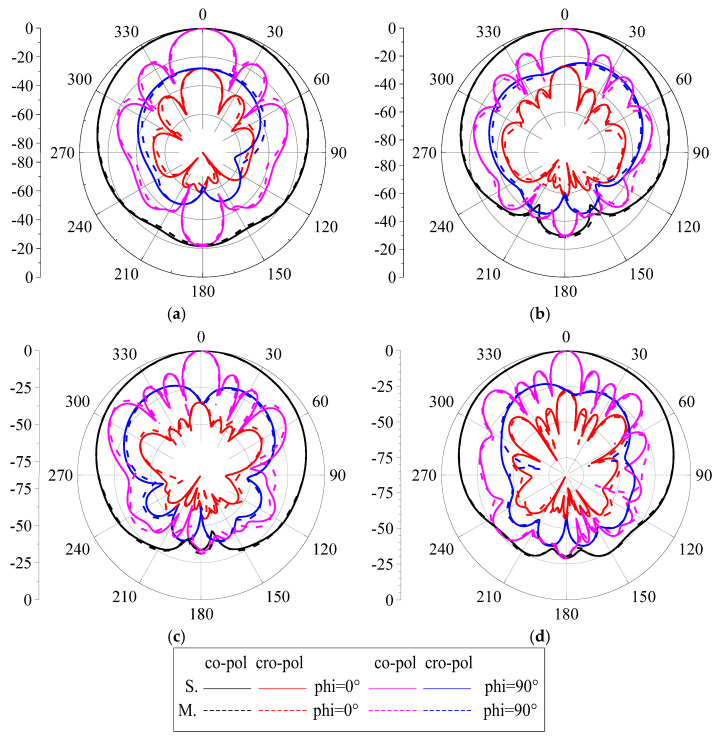
Simulated and measured radiation patterns of the proposed antenna array. (**a**) 1.7 GHz. (**b**) 2.4 GHz. (**c**) 3.2 GHz. (**d**) 3.7 GHz.

**Table 1 entropy-25-00274-t001:** Geometric parameters optimized for the proposed ME antenna.

Parameter	Value	Parameter	Value	Parameter	Value
*d*	26.0	*d* _2_	3.5	*f* _2*x*1_	6.4
*e_x_*	14.0	*r*	1.0	*f* _2*x*2_	3.6
*e_y_*	30.0	*d* _1_	8.0	*f* _2*z*1_	24.5
*H*	12.0	*f* _1*x*1_	6.4	*f* _2*z*2_	22.0
*m_y_*	10.0	*f* _1*x*2_	3.6	*f* _2*y*_	12.4
*m_z_*	30.0	*f* _1*z*1_	28.0	*R*	130.0
*d* _3_	10.0	*f* _1*z*2_	22.0	*h*	20.0
*p*	15.0	*f* _1*y*_	12.4	*d* _4_	12.5

**Table 2 entropy-25-00274-t002:** Comparison of the proposed antenna and references.

Ref.	Size (*λ*_0_^3^)	Bandwidth (%, GHz)	RL or VSWR	RE (%)	Gain (dBi)
[1]	0.68 × 0.68 × 0.17	76, 1.7–3.8	RL > 10 dB	>85	8.1
[2]	0.34 × 0.34 × 0.25	56, 1.62–2.87	VSWR < 1.5	>75	7.4
[3]	0.89 × 0.66 × 0.21	67, 1.39–2.8	RL > 15 dB	N.A.	>9
[9]	0.91 × 0.91 × 0.24	74, 1.7–2.7 and 3.3–3.7	RL > 10 dB	N.A.	>8.3
[10]	0.79 × 0.79 × 0.24	71.7, 1.7–2.7 and 3.4–3.6	<1.5	N.A.	8.1 ± 0.4, 6.6 ± 0.5
[15]	0.375 × 0.316 × 0.0016	9.6 (2.37–2.61), 28.8 (3.30–4.41), and 16.9 (4.98–5.90)	RL > 10 dB	>80	4.0
[16]	0.26 × 0.21 × 0.017	46.3, 3.12–5.0	RL > 10 dB	>70	2.5
Prop.	0.7 × 0.7 × 0.23	82.8, 1.62–3.91	RL > 15 dB	>90	>9.5

Return Loss (RL); Radiation Efficiency (RE).

## Data Availability

Not applicable.

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
