# Peer review of "A Novel Dual-Polarized Magnetoelectric Dipole Antenna and Its Array for LTE and 5G Sub-6 GHz Base Station Applications"

_entropy, 2023, doi:10.3390/e25020274_

Round 1
Reviewer 1 Report (New Reviewer)
The paper A Novel Dual-polarized Magnetoelectric Dipole Antenna and Its Array for LTE and 5G Sub-6 GHz Base Station Applications deals with designing and fabrication of a dual-polarized magneto-electric dipole antenna. The antenna is composed of L-shaped magnetic dipoles, planar electric dipoles, rectangular director, rectangular parasitic metal patches, and η-shaped feed probes. According to the simulations and measured results, the array of these antennas has good radiation characteristics, gain and broad bandwidth covering in LTE and 5G sub-6 GHz frequency band.
Generally, the overall organization of the paper is good. The article contains both theoretical as well as experimental sections. The length of a paper is appropriate. The paper is technically sound. The results of the proposed antenna and the array are promising.
However, the language level throughout the entire paper is somewhat low and needs to be increased significantly. I noticed a lot of various mistakes and language errors in the text. Therefore I recommend to perform extensive text editing and proof readings.
Author Response
Please see the attachment.

Reviewer 2 Report (New Reviewer)
A dual-polarized magneto-electric dipole antenna and its array is presented in this paper. The good performance of proposed antenna is obtained. The simulated results are good agreement with the measured results. I suggest to add one dual-polarized ME-dipole papers below in the references:
H. W. Lai, K. K. So, H. Wong, C. H. Chan and K. M. Luk, “Magnetoelectric dipole antennas with dual open-ended slots excitation,” IEEE Trans. Antennas Propag., vol. 64, no. 8, pp. 3338-3346, Aug. 2016.
Author Response
Please see the attachment.

Reviewer 3 Report (New Reviewer)
1. Kindly mention the full form of all standards in the manuscript.
2. Kindly comment on the work presented in Ref.[4] in the introduction section.
3. Kindly avoid mentioning the variable multiple times. For example λ0.
4. What do authors mean to say these “η-shape feeders”?
5. Kindly add a good fabricated photo of the antenna (Figure 15).
6. In the manuscript, the authors claim dual polarization. But I did not come across the axial ratio curve.
7. In Table II, also add a column containing dimensions in mm.
8. Kindly compare the following work, comment on them, and add to the comparison table:
a) Dual Polarized, Multiband Four-Port Decagon Shaped Flexible MIMO Antenna for Next Generation Wireless Applications," in IEEE Access, vol. 10, pp. 128132-128150, 2022, doi: 10.1109/ACCESS.2022.3227034.
b) Broadband and Compact Circularly Polarized MIMO Antenna With Concentric Rings and Oval Slots for 5G Application," in IEEE Access, vol. 10, pp. 29925-29936, 2022, doi: 10.1109/ACCESS.2022.3157914.
Round 2
Reviewer 1 Report (New Reviewer)
All revisions are ok, I recommend to accept the article.
Reviewer 3 Report (New Reviewer)
The authors have answered all the queries, hence my decision is to accept the paper.
This manuscript is a resubmission of an earlier submission. The following is a list of the peer review reports and author responses from that submission.
Round 1
Reviewer 1 Report
The paper requires minor modifications before publication. Specific comments follow:
1. The presentation of the antennas' configuration requires improvement. It is difficult for the reader to clearly understand how the antenna is structured. I would suggest remove the box in Fig. 1 and present the antenna in top and side views as well. Similarly in Fig. 4.
2. The antenna elements in the presented antenna array in Fig. 13 seem to be twited slightly for few degrees. Please clarify this issue.
3. English usage requires improvement, as well.
Author Response
Thank you for your hard work and effective suggestions on improving the quality of the paper. We have revised the corresponding contents of the paper item by item. Please see the attachment for details.

Reviewer 2 Report
Congratulations for the hard work.
Please check the English within text.
Author Response
Thank you for your hard work and affirmation of this paper. Based on your suggestions we have proofread the spelling mistakes of words in the paper word by word. We also have corrected the grammatical errors and sentences with unclear meaning in the paper sentence by sentence.
Reviewer 3 Report
(1) Several typo and grammar errors are observed in the manuscript.
(2) The novelty of this work is not clear and the performance compared with state-of-the-art is not superior.
(3) In Figure 15, simulated and measured lines can not be distinguished.
(4) In Table 2, the antenna size should be given in electrical dimensions.
Author Response

(The authors gave the same response as above.)

Round 2
Reviewer 1 Report
The authors addressed all the raised comments and clarified many parts. The manuscript has been substantially improved and deserves publication.